# Knock-Out of the Five Lysyl-Oxidase Family Genes Enables Identification of Lysyl-Oxidase Pro-Enzyme Regulated Genes

**DOI:** 10.3390/ijms231911322

**Published:** 2022-09-26

**Authors:** Tatyana Liburkin-Dan, Inbal Nir-Zvi, Hila Razon, Ofra Kessler, Gera Neufeld

**Affiliations:** Cancer Research Center, The Bruce Rappaport Faculty of Medicine, Technion-Israel Institute of Technology, Haifa 31096, Israel

**Keywords:** lysyl-oxidase, breast cancer

## Abstract

The five lysyl-oxidase genes share similar enzymatic activities and contribute to tumor progression. We have knocked out the five lysyl-oxidase genes in MDA-MB-231 breast cancer cells using CRISPR/Cas9 in order to identify genes that are regulated by LOX but not by other lysyl-oxidases and in order to study such genes in more mechanistic detail in the future. Re-expression of the full-length cDNA encoding LOX identified four genes whose expression was downregulated in the knock-out cells and rescued following LOX re-expression but not re-expression of other lysyl-oxidases. These were the AGR2, STOX2, DNAJB11 and DNAJC3 genes. AGR2 and STOX2 were previously identified as promoters of tumor progression. In addition, we identified several genes that were not downregulated in the knock-out cells but were strongly upregulated following LOX or LOXL3 re-expression. Some of these, such as the DERL3 gene, also promote tumor progression. There was very little proteolytic processing of the re-expressed LOX pro-enzyme in the MDA-MB-231 cells, while in the HEK293 cells, the LOX pro-enzyme was efficiently cleaved. We introduced point mutations into the known BMP-1 and ADAMTS2/14 cleavage sites of LOX. The BMP-1 mutant was secreted but not cleaved, while the LOX double mutant dmutLOX was not cleaved or secreted. However, even in the presence of the irreversible LOX inhibitor β-aminoproprionitrile (BAPN), these point-mutated LOX variants induced the expression of these genes, suggesting that the LOX pro-enzyme has hitherto unrecognized biological functions.

## 1. Introduction

The five genes of the lysyl-oxidase family (LOX, LOXL1-4) encode enzymes that catalyze the deamination of the ε-amino group of lysines of collagen and elastin monomers resulting in the formation of covalent cross-linkages and the stabilization of collagen and elastin fibers [1,2]. The catalytic domain of the lysyl-oxidases is highly conserved among lysyl-oxidases as is the amino-acids sequence of the lysyl tyrosyl quinone (LTQ) cofactor domain that is unique to these copper binding enzymes [1]. The lysyl-oxidase family members can be further divided into two sub-families. LOXL2, LOXL3 and LOXL4 are distinguished by the presence of four scavenger receptor cysteine-rich (SRCR) domains located at their N-termini, while LOX and LOXL1 lack such domains. Several lysyl-oxidase family members, notably LOX, LOXL2 and LOXL3 have been found to be upregulated by hypoxia and to enhance tumor metastasis [3,4,5]. The enhancement of tumor metastasis is accomplished using several mechanisms targeting intracellular as well as extracellular proteins. The expression of lysyl-oxidases in the tumor microenvironment results in enhanced extracellular matrix stiffness, promoting tumor cell invasiveness [6,7,8]. Lysyl-oxidases such as LOX and LOXL2 also produce hydrogen peroxide as a side product of their enzyme activity, and hydrogen peroxide in turn activates signal transduction via the focal adhesion kinase (FAK) pathway [9,10]. LOX, LOXL2 and LOXL3 were also found to regulate the availability of the SNAI1 and SNAI2 transcription factors, which function as promoters of epithelial-to-mesenchymal transition (EMT) due to their control of E-cadherin expression [5,11]. Lysyl-oxidases such as LOX and LOXL2 also function as pro-angiogenic factors that also contribute to tumor metastasis [12,13]. Lastly, it was found that some lysyl-oxidase family members such as LOX and LOXL2 can induce some biological activities such as EMT independently of their lysyl-oxidase enzyme activity [14,15,16].

Lysyl-oxidase (LOX) is the most intensively studied member of the lysyl-oxidase gene family. It is synthesized as a pro-enzyme (Pro-LOX) that is assumed to lack biological functions. Pro-LOX is cleaved following secretion by BMP-1 or ADAMTS2/14 to release the enzymatically active C-terminus, which represents the mature enzymatically active form of LOX (LOX) [17,18]. The cleavage also releases the N-terminal pre-pro fragment (LOX-PP). Interestingly, while mature LOX was found to enhance tumor progression, LOX-PP was found to function as a tumor suppressor [16,19,20]. Because several lysyl-oxidases are frequently expressed concomitantly by cells, it is unclear which properties of the lysyl-oxidases are shared between family members and which activities are unique to specific lysyl-oxidases. In order to identify biological functions that are unique to LOX in breast cancer cells, we have knocked out the five lysyl-oxidase genes in MDA-MB-231 breast cancer cells using CRISPR/Cas9. There have been a few other examples in which multiple genes belonging to a gene family have been knocked out using CRISPR/Cas9 in mammalian cells [21,22,23]. We chose to first knock out the genes encoding the LOX2-4 subfamily, followed by the knock out of the genes encoding the second lysyl-oxidase subfamily. This was accomplished by the sequential introduction of frame shift mutations into all alleles of these genes. This strategy enables the identification of functions specific to members of the gene family without interference by other related family members. These experiments resulted in the identification of several such genes whose expression is regulated by LOX but not by other lysyl-oxidase family members. Interestingly, activation of these genes by LOX occurs independently of the classical LOX enzyme activity and seems to be independent of the LOX pro-enzyme cleavage by the BMP-1 or ADAMTS2/14 proteases. In the future, we intend to study the mechanisms by which LOX regulates the expression of these genes and their possible biological roles in breast cancer tumor progression.

## 2. Results

In order to explore the biological properties of LOX without interference from other lysyl-oxidase family members, we knocked out the genes encoding the different lysyl-oxidases in MDA-MB-231 breast cancer cells by the introduction of frame shift mutations into the first or second exons of these genes using CRISPR/Cas9. The genes were knocked out sequentially in the order shown (Figure 1A). MDA-MB-231 cells are invasive cells derived from a triple negative breast cancer patient. We chose to knock out the lysyl-oxidase family genes in these cells because several lysyl-oxidases have been found to contribute to the progression of breast cancer [3,4,24,25]. These experiments resulted in the generation of several clones, in which all five lysyl-oxidase genes were knocked out (5× cells). After each knock-out cycle, we characterized the frame shift mutations that were introduced into each of the alleles of the lysyl-oxidase genes in the isolated clones of cells by DNA sequencing. The sequence changes in different alleles were determined manually and verified using the Tide online software package [26]. The frame shift mutations introduced into each of the alleles of the clone 6 5× cells, which we have subsequently used in many experiments, are shown in Figure 1C and Appendix A. Similar procedures were used to generate MDA-MB-231-derived clones of cells, in which we knocked out only the gene-encoding LOX (1× cells) (Figure 1B). The frame shift mutations introduced into each of the alleles of the clone 17 1× cells, which we have utilized subsequently in many experiments, are shown in Figure 1D and Appendix A. Interestingly, the migratory activity of the cells diminished progressively as the number of knocked-out lysyl-oxidase genes increased, reaching maximum inhibition following the knock out of the LOX gene (Figure 1E).

In order to identify the genes that were specifically regulated by LOX but not by other lysyl-oxidases, the mRNA expression profiles of the parental MDA-MB-231 cells and of several 5× knock-out cell clones derived from MDA-MB-231 cells, in which all five lysyl-oxidase genes were knocked out, were determined using deep mRNA sequencing (RNAseq). These mRNA expression profiles were compared with the mRNA profiles of these same 5× cells, in which we re-expressed the full-length unmodified LOX cDNA that did not contain any added epitope tags (Figure 2A). Interestingly, even though the LOX cDNA was efficiently expressed, we saw very little cleaved LOX in the conditioned medium of the parental MDA-MB-231 cells or in the conditioned medium of the 5x cells, in which we re-expressed the LOX cDNA (Figure 2A). We identified only four genes whose expression was downregulated following the knock out and whose expression was rescued following the re-expression of LOX (Figure 2C, arrow). This may be because the baseline expression level of LOX in the parental MDA-MB-231 cells was low, and these genes may therefore be particularly sensitive to LOX. These genes were the genes encoding STOX2, AGR2, DNAJC3 and DNAJB11.

The expression of none of these genes was previously associated with the expression of LOX. The storkhead box-2 (STOX2) gene encodes a transcription factor that had been associated with pre-eclampsia and oral squamous carcinomas [27,28]. It functions as a SMAD2/4 cofactor [29], and its expression is regulated by the TGF-β signaling pathway, which also regulates LOX expression, and vice versa [30,31]. Anterior gradient 2 (AGR2) is a disulfide isomerase that participates in protein folding in the endoplasmic reticulum [32]. The expression of AGR2 is also regulated by TGF-β [33], and its expression is associated with pancreatic cancer progression as well as with the progression of additional forms of cancer [34,35]. DnaJ Heat Shock Protein Family (Hsp40) Member C3 (DNAJC3) is also known as protein kinase inhibitor p58 (p58IPK) and is involved in the ER stress response. Its interaction with serine-threonine protein kinase RNA (PKR) is regulated by TGF-β [36]. Lastly, DNAJB11 is an endoplasmic reticulum heat shock protein that functions as a chaperon.

In order to verify that the expression of these genes in MDA-MB-231 cells is indeed regulated by LOX, we also re-expressed the full-length LOX cDNA without any epitope tags in the clone 17 1× knock-out cells (Figure 2B). In the conditioned medium of these cells, we also detected only residual amounts of cleaved LOX (Figure 2B). We then compared the effects of the knock out and the re-expression of LOX on the expression of the mRNAs encoding the four LOX-regulated genes that we identified in the clone 6 5× knock-out cells using quantitative reverse PCR (qRT-PCR). Indeed, the expression of the STOX2, AGR2 and DNAJB11 mRNAs was inhibited significantly in the clone 6 5× cells as well as in the clone 17 1× cells, and the expression of these genes was rescued following the re-expression of the LOX cDNA in the clone 6 5× cells as well as in the clone 17 1× cells (Figure 3). In contrast with the result obtained for the 5× knock-out cells, which agreed with the results obtained in the RNAseq screen, the expression of DNAJC3 was not inhibited in the clone 17 1× knock-out cells. However, re-expression of LOX enhanced DNAJC3 expression in these cells as well (Figure 3). To find out if other lysyl-oxidases could affect the expression of these four genes, we also expressed in the clone 6 5× cells the full-length cDNAs encoding LOXL2, LOXL3 and LOXL4. However, the expression of the STOX2, AGR2, DNAJB11 and DNAJC3 genes could not be rescued following the expression of these three lysyl-oxidases. Taken together, these results suggest that the expression of these four genes is specific to LOX and that the other lysyl-oxidases (with the exception of the LOXL1 gene, which we did not test) are not able to regulate their expression in these cells (Figure 3).

When we re-expressed LOX in the 5× clone 6 knock-out cells, we identified additional genes that were significantly upregulated following LOX re-expression (Figure 4A). However, the expression of these genes was not downregulated in the 5× knock-out cells. We verified, using qRT-PCR, that the expression of three of these genes is indeed regulated by LOX in both the 5× clone 6 and in 1× clone 17 knock-out cells (Figure 4B and Figure 8). These were the HSP90B1, DERL3 and HSPA5 genes. The HSP90B1 (endoplasmin) gene encodes a heat shock chaperone, which was reported to interfere with TGF-β signaling [37], and is associated with the progression of multiple myeloma [38], breast cancer [39] and osteosarcoma [40], to mention a few examples. The derlin-3 (DERL3) gene’s expression is associated with the progression of breast cancer [41], and the HSPA5 heat shock protein or BIP has also been implicated in cancer [39] (Figure 4B). The expression of these genes was not induced by either LOXL2 or LOXL4 (Figure 4B). However, the expression of both HSP90B1 and DERL3 was also induced by LOXL3. The expression of HSPA5 also seems to be induced by LOXL3, although in this case, it did not reach statistical significance (Figure 4B). LOX re-expression also enhanced the expression of these genes in the 1× clone 17 knock-out cells (Figure 8). It is possible that these genes were not downregulated in the 5× knock-out cells because the basal expression level of LOX and LOXL3 in the MDA-MB-231 cells may not be sufficient to induce their expression.

Secreted LOX is cleaved by the BMP-1 and ADAMTS2/14 proteases. The ~30-kDa and ~25-kDa C-terminals that are generated following cleavage by these proteases represent the mature, enzymatically active form of LOX [17,18,20]. However, we could detect very little if any cleaved ~25–30-kDa LOX in the conditioned medium of the 5× or 1× knock-out cells in which we re-expressed the LOX cDNA (Figure 2A,B). These observations suggest that the LOX pro-enzyme, or the LOX peptides generated by proteases other than BMP-1 or ADAMTS2/14, may be responsible for the upregulation of these LOX-regulated genes in MDA-MB-231 cells. To examine this possibility, we introduced point mutations into the BMP-1 and into the ADAMTS2/14 cleavage sites of the LOX cDNA (Figure 5A) to generate full-length LOX cDNAs containing both mutations (DmutLOX) as well as cDNAs containing only the BMP-1 (BMP-mut LOX) or ADAMTS2/14 (ADAMTS-mut LOX) cleavage site mutations. The BMP-1 cleavage site mutation was performed following a previous publication [17]. We also added a myc epitope tag upstream of the LOX stop codon of the unmodified LOX as well as to the point-mutated LOX variants in order to be able to distinguish the products of these constructs from endogenous LOX. To determine the effects of these mutations on the secretion and cleavage of LOX, we first expressed the LOX/myc cDNA as well as the cDNAs encoding the point-mutated LOX/myc variants in HEK293 cells, as in these cells, LOX is efficiently cleaved following secretion [17]. Indeed, following the expression of the LOX/myc cDNA in these cells, the conditioned medium contained roughly equal concentrations of uncleaved LOX and mature, ~35-kDa cleaved, myc-tagged LOX (Figure 5C). The cDNAs encoding the point-mutated forms of LOX were expressed in these cells at comparable levels of expression as determined by examination of the cell lysates (Figure 5B). BMP-mut LOX was efficiently secreted but, as expected, failed to be cleaved, suggesting that under these conditions, there is no cleavage at the ADAMTS2/14 cleavage site. In contrast, ADAMTS-mut LOX was poorly secreted, and the fraction that was secreted was partially cleaved by BMP-1 (Figure 5C). Finally, DmutLOX failed to be cleaved and was secreted very poorly (Figure 5C). Interestingly, in the lysates derived from these cells, we also found additional myc-tagged bands which were likely generated from LOX by uncharacterized proteases, as these bands were not present in the cell lysates derived from wild-type HEK293 cells (Figure 5B).

We then expressed the myc-tagged LOX and the various myc-tagged LOX point mutants in the clone 6 5× knock-out cells and clone-17 1× knock-out cells. Both LOX and the LOX mutants were efficiently expressed in the clone 6 cells with similar efficiency (Figure 6A). The cell lysates derived from these cells also contained lower molecular weight bands. These bands probably represent additional myc-tagged, cleaved, LOX-derived peptides produced by uncharacterized proteases since they were not detected in the clone 6 5× knock-out cells (Figure 6A). The conditioned medium of the clone 6 cells expressing the wild-type LOX cDNA contained the full-length uncleaved LOX pro-enzyme and very little cleaved mature LOX (Figure 6B arrow). BMP-mut LOX was secreted but, as expected, was not cleaved (Figure 6B), and when the conditioned medium was assayed for enzyme activity using the Amplex red assay, we could not detect any significant enzyme activity (Appendix A). ADAMTSmut LOX was not secreted, and we could not detect cleavage products corresponding to the mature LOX forms, possibly because so little was secreted. DmutLOX was not secreted, and in this case as well, we could not detect in the conditioned medium of the cells C-terminal fragments corresponding to the expected mass of mature LOX (Figure 6B). We also conducted an analogous experiment using the clone 17 1× knock-out cells. All the LOX variants were efficiently expressed, and in these cells, we also observed cleaved peptides that contained the myc epitope tag which were likely generated by uncharacterized proteases (Figure 6C). In these cells too, wild-type LOX was efficiently secreted, but we could not detect any cleaved mature LOX (Figure 6D). In the conditioned medium of these cells, we observed a 35-kDa band that was strongly stained with the anti-myc antibody. This band was also present in the clone 17 knock-out cells that did not express LOX, and this probably represents a nonspecific staining. In the conditioned medium of the clone 17 cells that expressed the LOX mutants, we observed the same behavior as in the clone 6 cells. BMP-mut LOX was efficiently secreted but was not cleaved, while ADAMTS-mut LOX and DmutLOX were not secreted and were not cleaved (Figure 6D).

We then determined if the expression of STOX2, AGR2, DNAJB11 and DNAJC3 genes could also be induced by LOX mutants that did not undergo cleavage by BMP-1 or ADAMTS2/14. Indeed, we found that the expression of these genes could be induced by DmutLOX as well as by ADAMTS-mut LOX and BMP-mut LOX when the cDNAs encoding them were expressed in either the clone 6 5× knock-out cells or clone 17 1× knock-out cells (Figure 7). The only exception was the AGR2 gene, which was only partially induced by the double mutant and by the single mutants in the clone 6 cells. However, the induction failed to reach statistical significance in the case of the single mutants. Notably, these single mutants did enable significant induction of AGR2 expression in the clone 17 1× cells (Figure 7). We similarly tested three of the genes that were not downregulated by the knock-out of LOX but whose expression was induced following the re-expression of LOX in the clone 6 5× knock-out cells (Figure 4). The HSP90B1, DERL3 and HSPA5 genes were also induced by the three LOX point mutant variants in the 5× and 1× knock-out cells (Figure 8), suggesting that the expression of these three genes is also induced by the LOX pro-enzyme rather than by cleaved, enzymatically active LOX (Figure 8).

To make sure that the enzyme activity of LOX is indeed not required for the upregulation of these genes, we determined if the induction of the expression of the STOX2, DNAJC3 and HSP90B1 genes by LOX was inhibited by BAPN [42]. In agreement with the results obtained using the cleavage mutants, we found that the expression of these genes was induced by wild-type LOX even in the presence of high BAPN concentrations (Figure 9). Taken together, these observations suggest that the LOX pro-enzyme has hitherto unsuspected biological functions that do not require processing by the ADAMTS2/14 or BMP-1 proteases at the classical cleavage sites of LOX.

## 3. Discussion

Lysyl-oxidase was identified as a secreted enzyme that catalyzed the formation of covalent bonds between collagen and elastin monomers. It was synthesized as a pro-enzyme precursor and was cleaved by the BMP-1 or ATAMTS2/14 proteases following its secretion into the extracellular space. The resulting C-terminal peptides represent the enzymatically active mature LOX [17,18]. Cleavage by the BMP-1 protease also released the catalytically inactive N-terminal LOX-PP peptide, which functions as an inhibitor of the ras signal transduction pathway and as a tumor suppressor [19,20]. LOX is a member of the lysyl-oxidase gene family, which includes five different proteins that share a highly homologous C-terminal and very similar enzyme activities [43]. Under hypoxic conditions, LOX is overexpressed and promotes the metastasis and invasion of breast cancer cells [9,44,45]. In order to identify functions unique to LOX in breast cancer cells, we knocked out the five members of the lysyl-oxidase gene family in MDA-MB-231 breast cancer-derived cells.

We chose to knock out the different lysyl-oxidase genes sequentially, one after the other, making sure after each step that all alleles were disrupted. The drawback of this strategy is that there is also an accumulation of off-target effects which cannot be readily controlled. Therefore, since the main focus here was LOX, we concentrated only on the identification of genes whose expression was changed following the knock out operation and that were rescued following LOX re-expression but not following the re-expression of other lysyl-oxidase family members. We then verified that the expression of these genes was controlled by LOX in a separate knock-out model, in which we only knocked out the LOX gene but not the genes encoding other lysyl-oxidases, since in this model, there should be fewer off-target effects, and therefore, if we obtained similar results in this model, it would strengthen confidence in the results obtained using cells in which we knocked out all five lysyl-oxidase genes. The lysyl-oxidase genes were divided into two subfamilies. We chose to knock out the first subfamily (LOXL2, LOXL3 and LOXL4) followed by the second group of genes (LOX and LOXL1). It remains to be determined if the order of the knock outs can affect the results or not. This will need to be determined in further experiments.

CRISPR/Cas9 was used in a few other studies to knock out multiple genes in mammalian cells. We have previously knocked out the genes that comprise the neuropilin receptor family [21]. However, this represented a much easier technical challenge as only two genes make up this gene family. In another study, 3 out of the 10 Frizzled family receptors were knocked out [22], and in yet another study, several genes that were suspected to have roles in maternal-to-zygotic transition were knocked out [23]. These last two studies used a strategy different than the one we used here, as the genes were not knocked out sequentially but simultaneously. Several different guide RNA species were used in these studies as well as in the present study to knock out the target genes. In these studies, as well as in the present study, the use of multiple guide RNA species is likely to be associated with an increased incidence of off-target events. It is difficult to determine how such off-target events may affect the behavior of the cells. Nevertheless, despite these concerns, we are not aware of another strategy that enables the unbiased identification of genes whose expression is regulated by a member of a gene family but not by other related family members.

LOX was implicated previously in the control of gene expression [11,45,46]. We therefore screened MDA-MB-231 breast cancer cells in which we knocked out all five lysyl-oxidase genes (5× cells) for the genes whose expression was either inhibited or enhanced and then further selected genes whose expression was rescued following the expression of the full-length LOX cDNA. These experiments resulted in the identification of the STOX2, AGR2, DNAJC3 and DNAJB11 as genes whose expression was downregulated following the knock out of the five lysyl-oxidase genes and whose expression could be rescued following LOX re-expression but not by other lysyl-oxidases. Similar observations were made using MDA-MB-231 cells in which we only knocked out the LOX gene. Interestingly, none of these genes have been previously identified as genes that are regulated by LOX. Notably, the expression of STOX2 and AGR2 had been previously linked to tumor progression [28,32,47].

All of these genes except STOX2, which is a transcription factor, are located in the endoplasmic reticulum and participate in the response to stress and protein misfolding response. However, these may not be the only functions of these genes. Since STOX2, AGR2, DNAJC3 and DNAJB11 were significantly downregulated following the LOX knock out, it is likely that their upregulation by LOX is not the result of nonspecific upregulation due to LOX overexpression. Overexpression of other lysyl-oxidases failed to rescue the expression of these genes, further suggesting that their upregulation by LOX is not a nonspecific response to high protein expression levels.

Interestingly, even when we overexpressed the LOX cDNA in the MDA-MB-231 cells, we could detect very little mature cleaved LOX in their conditioned medium. This was in sharp contrast to the HEK293 cells, which contained a high concentration of cleaved active LOX in their conditioned medium following LOX overexpression. These observations suggest that the activation of the LOX-regulated genes that we identified may not require prior cleavage of LOX by either BMP-1 or ADAMTS2/14. To determine whether cleavage by these proteases is required for LOX-induced activation of these genes, we generated LOX variants in which we point mutated the cleavage sites of BMP-1 and ADAMTS2/14. When expressed in HEK293 or in MDA-MB-231 cells, the BMP-1 cleavage site mutant was secreted but failed to be cleaved, while the ADAMTS2/14 cleavage site mutant was secreted very poorly. The double mutant DmutLOX was also very poorly secreted, if at all, and failed to be cleaved. However, when re-expressed in the 5x knock-out MDA-MB-231 cells, these three mutated LOX variants were still able to enhance the expression of the STOX2, AGR2, DNAJB11 and DNAJC3 genes as well as wild-type LOX. This was also true for additional genes such as the DERL3, HSP90B1 and HASPA5 genes, which we also found to be regulated by LOX in MDA-MB-231 cells. Furthermore, the classical oxidase activity of LOX was not required for the activation of these genes by LOX since their expression was induced by LOX even in the presence of high concentrations of the LOX irreversible inhibitor BAPN.

To conclude, our experiments have identified in MDA-MB-231 breast cancer cells several LOX-regulated genes. The expression of four of these genes seems to be induced by LOX but not by other lysyl-oxidases. Furthermore, the classical lysyl-oxidase enzyme activity of LOX is apparently not required for this activity, nor is secretion of the LOX pro-enzyme or cleavage of the LOX pro-enzyme by either BMP-1 or ADAMTS2/14 required. These observations suggest that contrary to previous assumptions, the LOX pro-enzyme may possess independent biological functions, some of which may be associated with its pro-tumorigenic activity. It is also possible that peptides generated from the LOX pro-enzyme by proteases other than BMP-1 or ADAMTS2/14 are responsible for the activation of the genes that we found to be regulated by LOX in these cells. STOX2 is a transcription factor, and it was the only LOX upregulated gene we identified that is not known to be involved in the response to stress. We will therefore concentrate our future efforts on the elucidation of the mechanism by which LOX induces expression of the STOX2 gene and on the elucidation of the role of STOX2 in the biological responses to LOX expression in breast cancer cells and normal cells.

## 4. Materials and Methods

### 4.1. Antibodies and Reagents

The anti-C-*myc* mouse monoclonal (9E10) antibodies and mouse anti-LOXL1 antibodies (sc-166632) were from Santa Cruz, Biotechnology, Inc., Dallas, TX, USA. The rabbit anti-LOX (ab31238) and rabbit anti-LOXL4 (ab88186) were from Abcam (Cambridge, UK). The mouse anti-actin (clone AC-74, A5316), goat anti-rabbit IgG peroxidase conjugate (A6154) and goat anti-mouse IgG peroxidase conjugate (A4416) and puromycin were from Sigma (Sigma-Aldrich Israel Ltd., an affiliate of Merck KGaA, Darmstadt, Germany). The rabbit polyclonal anti-LOXL2 and rabbit polyclonal anti-LOXL3 were previously produced in our lab [3,48]. The basement membrane matrix (Cultrex Basement Membrane Extract, PathClear, #3432-005-01) was from Bio-Techne (R&D Systems, Inc., Minneapolis, MN, USA). The DMEM-high glucose (Dulbeco’s modified eagle’s medium, 01-055-1A), DMEM-high glucose without phenol red (01-053-1A) and fetal calf serum were purchased from Biological Industries LTD. (Biological Industries Israel Beit-Haemek, Kibbutz Beit-Haemek, Israel). The T4 DNA ligase (M180A) was from Promega (Promega Corporation, Madison, WI, USA). The NEBuilder HiFi DNA Assembly Master Mix (E2621), Phusion High-Fidelity DNA Polymerase (M0530) and the different restriction enzymes were from New England Biolabs (MA, Ipswich, UK). Lipofectamine 3000 was purchased from Invitrogen (Cat. No. L3000008. Thermo Fisher Scientific corporation, Waltham, MA, USA). Lipofectamine 3000 was purchased from Invitrogen (Cat. No. L3000008. Thermo Fisher Scientific corporation, Waltham, MA, USA). The following kits were from Macherey-Nagel (Düren, Germany): NucleoSpin RNA Plus (Cat. No. 740984.50), NucleoBond Xtra Maxi Plus (740416.10), NucleoSpin Plasmid EasyPure (740727.250) and NucleoSpin Tissue (740952.50). The qScript cDNA Synthesis Kit was from Quantabio (Beverly, MA, USA) (Cat. No. 95047-100), and the REDExtract-N-Amp^TM^ Tissue PCR Kit (XNAT-100RXN) was from Sigma (Sigma-Aldrich Israel Ltd., an affiliate of Merck KGaA, Darmstadt, Germany, Rehovot, Israel).

### 4.2. Plasmids

The NSPI-CMV-MCS-myc-His lentiviral expression vector was previously described [3]. The pSpCas9(BB)-2A-GFP (PX458) plasmid was from Addgene (Watertown, MA, USA) (deposited by Feng Zhang [49]). The pENTR1A-GFP-N2 (#19364) plasmid was from Addgene (Watertown, MA, USA) (deposited by Eric Campeau [50]). The pLenti6/V5-DEST plasmids was purchased from Invitrogen (Thermo Fisher Scientific corporation, Waltham, Massachusetts, USA). The ∆NRF (pCMV dR 8.74) and pMD2-VSV-G vectors for lentivirus production were kindly provided by Dr. Tal Kafri (University of North Carolina at Chapel Hill, NC, USA).

### 4.3. Cell Lines

HEK293 cells were purchased from the American Type Culture Collection (ATCC) (Manassas, VA, USA) and cultured as previously described [51]. HEK293-FT cells were purchased from Invitrogen (Thermo Fisher Scientific corporation, Waltham, MA, USA) and cultured as described for HEK293 cells. MDA-MB-231 cells were kindly provided by Dr. Michael Klagsbrun (Harvard University, Boston, MA, USA) and cultured as described for the HEK293 cells. Puromycin (2 µg/mL, Sigma-Aldrich Israel Ltd., an affiliate of Merck KGaA, Darmstadt, Germany, Rehovot, Israel) was used to select the infected cells. The XL-1-Blue *E. coli* competent cells were prepared in our laboratory. The JM109 *E. coli* strains were purchased from Promega(Promega Corporation, Madison, WI, USA). HIT DH5α competent cells (RBC-RH618) were purchased from Real Biotech Corporation (Banqiao City, Taipei County, Taiwan). All competent cell types were used for transformation by heat shock according to the manufacturer’s instructions.

### 4.4. Generation of MDA-MB 231 5× Knock-Out Cells Using CRISPR/Cas9

MDA-MB-231 cells were transfected with pSpCas9(BB)-2A-GFP plasmids containing a guide RNA sequence either for LOX (for LOX, two sgRNA pairs were used simultaneously) or all the other lysyl-oxidase family members [52]. All the sgRNAs (Appendix A) were chosen using the CRISPR Design Tool (http://crispr.mit.edu/). Fluorescence-activated cell sorting (FACS) was performed 48 h after the transfection for selection of the GFP-expressing cells. The sorted cells were submitted to limiting dilution cloning. The resulting clones were tested by sequencing to detect frame shift knock-out clones of the cells. Each sequence was compared to the wild-type sequence of the gene and thoroughly examined in order to find the ones with insertion or deletion mutations causing frame shift disruption in both alleles [52], both manually and also by using an online tool (http://shinyapps.datacurators.nl/tide/) [26,53]. Clones which were found to have frame shift mutations in both alleles were subjected to qRT-PCR analysis for confirmation of decreased RNA levels. After verification, one knock-out clone was chosen and carried on to the next lysyl-oxidase family member knock-out, which was performed using the same procedure. The lysyl-oxidase family genes were knocked out sequentially in the order shown (Figure 1A) to generate 5× knock-out cells. In addition, we also knocked out only the LOX gene in the MDA-MB-231 cells using a similar procedure to generate the 1× LOX knock-out cells.

### 4.5. Cell Invasion

A 96-well ImageLock plate was coated with a layer of 100 μg/mL of the basement membrane matrix and incubated overnight at 37 °C. The cells were seeded at the optimized cell density (30 K cells/well) and incubated overnight at 37 °C. A top layer of 3 mg/mL of the basement membrane matrix was prepared separately in a 96-well plate and stored in a cool box (4 °C, Biocision, Corning^®^ CoolBox™. (Sigma-Aldrich Israel Ltd., an affiliate of Merck KGaA, Darmstadt, Germany, Rehovot, Israel).) in order to maintain a low temperature. The WoundMaker™ procedure under the manufacturer’s instructions was performed to create precise and reproducible wounds in all the plate wells. The cells were washed with 100 μL of cold medium to remove debris and to prevent dislodged cells from settling and reattaching. Then, the cell plate was cooled in a cool box for 5 min. Next, the medium was aspirated, and the cells were overlaid with the basement membrane matrix’s top layer, which was in the 96-well plate in the cool box. To gel the top layer, the cell plate was warmed to 37 °C by placing it in the cell incubator (37 °C). After 3 min, an additional 100 μL per well of warm medium was added and the assay plate, which was placed into the IncuCyte ZOOM^®^ instrument (HD/2CLR, Essen BioScience, Sartorius, Göttingen, Germany). The scanning schedule was set to repeat scanning every 45 min for 72 h. Scanning was conducted using 10× objectives. After 35 h, the MDA-MB-231 parental cells managed to close the scratch area gap. Therefore, this time point was used as the end point for the analysis. Each assay was performed using six replicates for each cell line.

### 4.6. Expression of Recombinant Lysyl-Oxidases

The full-length cDNAs of the different lysyl-oxidases, except for LOXL4, were cloned into the NSPI-CMV-MCS-myc-His lentiviral expression vector without any epitope tags using the NEBuilder HiFi DNA Assembly according to the instructions of the manufacturer (New England Biolabs, Ipswich, MA, USA). The full-length cDNA of LOXL4 without any epitope tags was cloned into the gateway entry vector pENTR1A-GFP-N2 using the NEBuilder HiFi DNA Assembly. LOXL4 was then transferred by recombination into the pLenti6/V5-DEST lentiviral expression vector according to the instructions of the manufacturer (Invitrogen). To generate the LOX mutants, we used a custom-made plasmid containing a segment of the LOX gene, with point mutations in the BMP-1 and ADAMTS2/14 cleavage sites (Figure 5A) from Syntezza Bioscience Ltd. (Jerusalem, Israel). The point mutations to the BMP-1 site were described in a previous publication [17]. To generate the DmutLOX construct that contained both mutated cleavage sites, the relevant segment from the custom-made plasmid was assembled with the complementing segments derived from the wild-type LOX gene sequence and the NSPI-CMV-MCS-myc-His plasmid, framed with the Myc-His tag, using the NEBuilder HiFi DNA Assembly. The same procedure was used to generate LOX constructs with the single mutations (BMPmutLOX and ADAmutLOX). To generate those, only the mutated BMP-1 cleavage site segment or the mutated ADAMTS2/14 cleavage site segment was used. The primers used are detailed in Appendix A. Production of the lentiviruses using these plasmids and stable infection of the target cells (MDA-MB-231 and HEK239) was performed essentially as previously described [54].

### 4.7. Next-Generation RNA Sequencing

RNA replicates of a high RNA integrity (RIN ≥ 8) were processed for RNA-Seq at the Crown Institute for Genomics (G-INCPM, Weizmann Institute of Science, Rehovot, Israel), where 500 ng of total RNA for each sample was processed using the in-house poly-A-based RNA-Seq protocol (INCPM mRNA Seq). The libraries were evaluated by Qubit and TapeStation. The sequencing libraries were constructed with barcodes to allow multiplexing of all samples on one lane of an Illumina NextSeq machine using the Single-Read 60 protocol (v4). The output was ~21 million reads per sample. Poly-A/T stretches and Illumina adapters were trimmed from the reads using cutadapt [55]. The resulting reads shorter than 30 bp were discarded. The reads for each sample were aligned independently to the *Homo sapiens* reference genome GRCh38 using STAR [56] and supplied with gene annotations downloaded from Ensembl (and with EndToEnd option). The percentage of the reads that were aligned uniquely to the genome was ~85%. Counting proceeded over the genes annotated in Ensembel release 83 using htseq-count [57]. Only the uniquely mapped reads were used to determine the number of reads falling into each gene (intersection-strict mode). Differential analysis was performed using DESeq2 package [58] with the betaPrior, cooksCutoff and independent filtering parameters set to 51 False. The raw *p* values were adjusted for multiple tests using the procedure of Benjamini and Hochberg [59]. Differentially expressed genes were determined by a p-adj of <0.05, absolute fold changes >2 and max raw counts >30. Heatmap plotting was performed using the ComplexHeatmap package (GitHub: https://github.com/jokergoo/ComplexHeatmap (R)) on the variance stabilizing transformation counts of DESeq2 of each gene.

### 4.8. Quantitative Real-Time PCR

Quantitative real-time PCR was performed using a StepOne Plus Real-Time PCR System (Applied biosystems, Waltham, MA, USA) with the TaqMen Universal PCR Master Mix according to the instructions of the manufacturer (Applied biosystems, Waltham, MA, USA). Assays for each gene target were performed in triplicate for all cDNA samples. The normalizing gene was RPLPO. The data were analyzed by the software StepOne (Applied biosystems, Waltham, MA, USA) using the relative quantitation-comparative CT method. The primers used are listed in Appendix A.

### 4.9. Generation of Concentrated Conditioned Media and LOX Amplex Red Activity Assays

See the Appendix A.

### 4.10. Statistical Analysis

Statistical analysis was carried out using GraphPad Prism 5 software. Statistical significance was determined using the Mann–Whitney one-tailed nonparametric test or student’s *t* test with Welch’s correction. Data were obtained from at least three independent experiments performed in triplicate unless otherwise stated. The data are represented as the mean ± SEM. The following designations were used: * *p* < 0.05, ** *p* < 0.01 and *** *p* < 0.001, as well as nonspecific (ns).

## Figures and Tables

**Figure 1 ijms-23-11322-f001:**
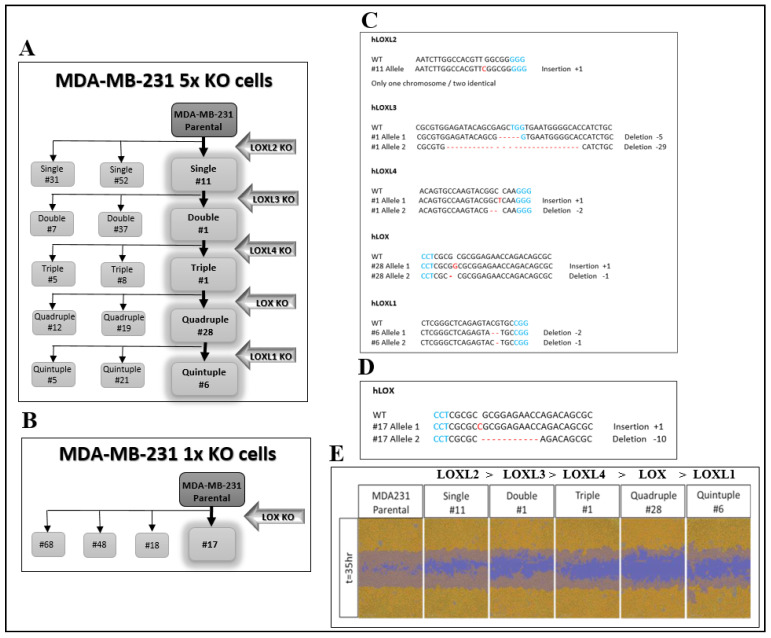
CRISPR/Cas9-mediated knock out of the five LOX family genes in MDA-MB-231 breast cancer cells. (**A**) A schematic of the sequential knock out of the lysyl-oxidase family genes that resulted in the generation of the 5× knock-out clones (clones 5, 6 and 21). Knock out was performed using CRISPR/Cas9-mediated introduction of frame shift mutations into the first encoding exon of each gene. (**B**). The 1× LOX single knock-out clones were generated similarly. (**C**) The frame shift mutations introduced into each of the alleles of the various lysyl-oxidase family genes in the generation of the clone 6 5× knock-out cells. PAM sequences are highlighted in blue. Insertions or deletions are highlighted in red. (**D**) The frame shift mutations introduced into each of the LOX alleles of clone 17 1× knock-out cells. (**E**) The invasiveness of MDA-MB-231 cells as well as cells from each of the intermediate sequential knock-out clones generated on the route to the final clone 6 5× knock-out cells was evaluated as described in Materials and Methods. The assay was repeated independently three times with similar results.

**Figure 2 ijms-23-11322-f002:**
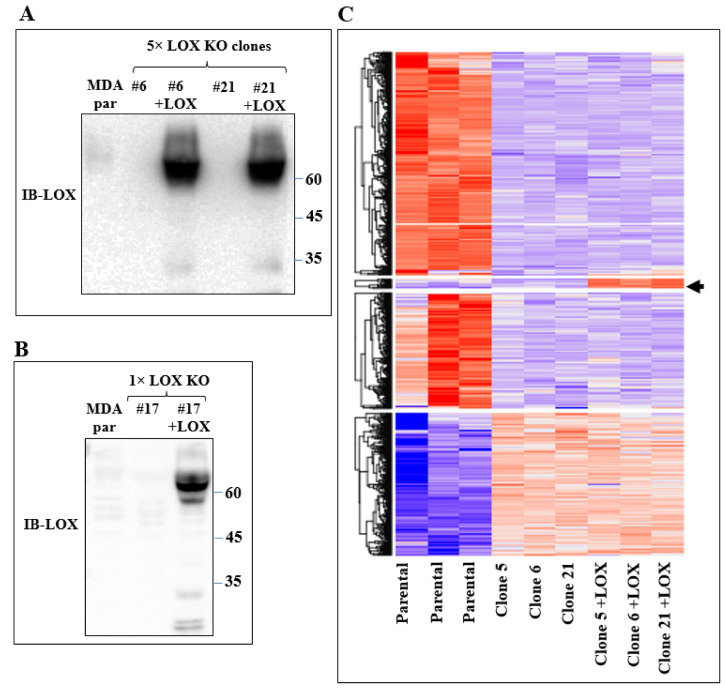
Re-expression of LOX in 5× knock-out clones resulted in changes in gene expression. (**A**) Western blot analysis using anti-LOX antibodies of equal volumes of conditioned medium of confluent MDA-MB-231 cells (MDA par), clone 6 and clone 21 cells and of these cells following LOX cDNA re-expression. (**B**) Western blot analysis using anti-LOX antibodies of equal volumes of conditioned medium of confluent MDA-MB-231 cells (MDA par) and clone 17 cells and of these cells following LOX cDNA re-expression. (**C**) Heat map showing color-coded expression levels of differentially expressed genes in three different plates of parental MDA-MB-231 cells, in three different 5× knock-out clones (clones 5, 6 and 21) and in these knock-out clones after re-expression of full-length LOX cDNA. The LOX cDNA did not contain epitope tags. Arrow points to a group of genes whose expression was either unchanged or inhibited following the 5× knock out but whose expression was upregulated following re-expression of the LOX cDNA.

**Figure 3 ijms-23-11322-f003:**
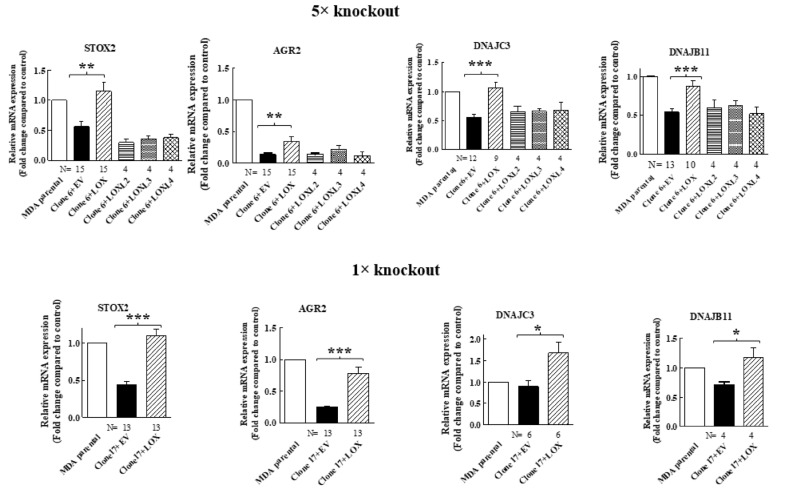
The expression of STOX2, AGR2, DNAJB11 and DNAJC3 is rescued by LOX re-expression but not by re-expression of other lysyl-oxidases, with qRT-PCR analysis of the effects of the re-expressed lysyl-oxidases in 5× clone 6 and 1× clone 17 knock-out cells on the expression of the indicated genes. No epitope tags were added to the cDNAs encoding the different lysyl-oxidases. Results were normalized with the expression level in MB-MDA-231 parental cells in each of the experiments. Data represent the mean ± SEM of N independent experiments. Statistical significance was evaluated using unpaired student’s *t* test with Welch’s correction. * *p* < 0.05, ** *p* < 0.01, *** *p* < 0.001.

**Figure 4 ijms-23-11322-f004:**
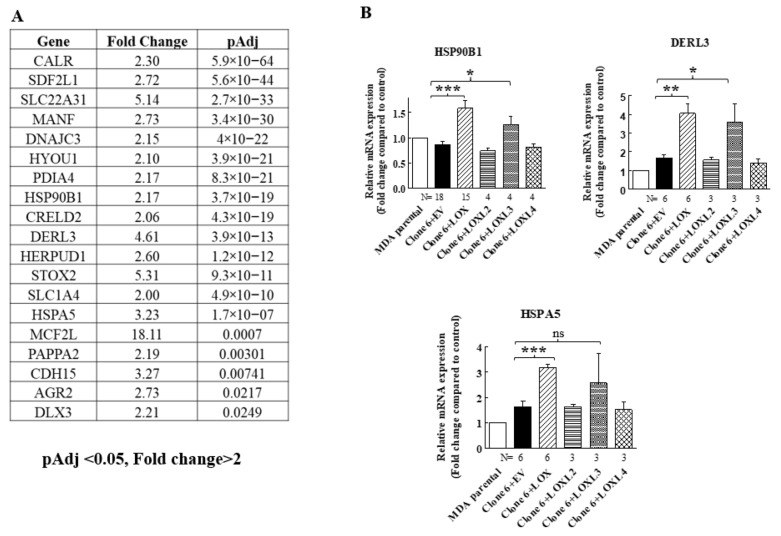
LOX expression in MB-MDA-231 5× knock-out cells resulted in the upregulation of the expression of the HSP90B1, DERL3 and HSPA5 genes. (**A**) A list of genes whose expression was significantly upregulated in an RNAseq screen following LOX re-expression in three different MB-MDA-231 5× knock-out clones. Listed are the genes whose expression was enhanced at least twofold and whose adjusted *p* values were lower than 0.05 following LOX re-expression. (**B**) The expression levels of HSP90B1, DERL3 and HSPA5 were determined using qRT-PCR in MB-MDA-231 clone 6 5× knock-out cells before and after re-expression of the indicated lysyl-oxidases. Results were normalized in comparison to the expression levels of the genes in parental MDA-MB-231 cells. Data represent the mean ± SEM of N independent experiments. Statistical significance was evaluated using unpaired student’s *t* test with Welch’s correction. * *p* < 0.05, ** *p* < 0.01, *** *p* < 0.001.

**Figure 5 ijms-23-11322-f005:**
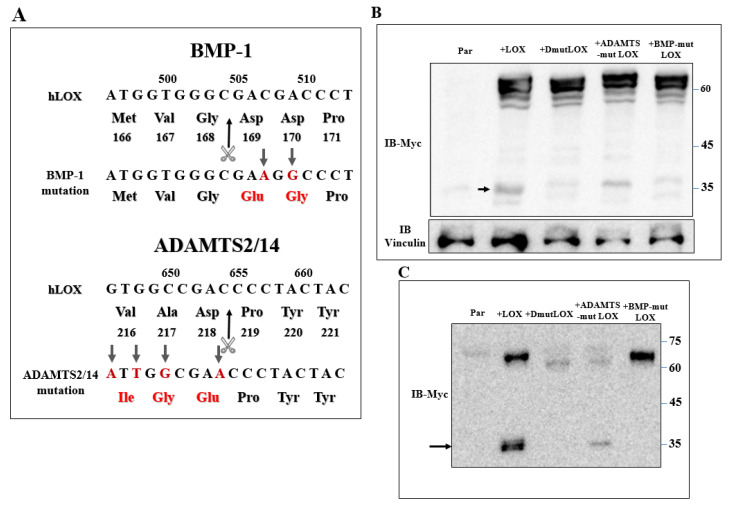
Generation of point-mutated LOX constructs. (**A**) A schematic illustration of the point mutations introduced into the LOX cDNA and protein sequence to generate the BMP-1 cleavage site mutant (BMPmut LOX), the ADAMTS2/14 cleavage site mutant (ADAMTSmut LOX) and the double mutant (DmutLOX) that contains both mutations. A myc epitope tag was added upstream of the stop codons of all constructs. Changed nucleotides and amino acids are indicated in red. The scissors mark the corresponding cleavage sites. Numbers above nucleotide sequences and under protein sequences indicate nucleotide numbers and amino-acid numbers, respectively. (**B**) Western blot analysis using anti-myc antibodies. LOX expression was assayed in cell lysates of HEK293 cells and HEK293 cells in which LOX and the various LOX mutants were expressed. Vinculin was used as a loading control. Cleaved LOX is marked by an arrow. (**C**) Western blot analysis using anti-myc antibodies of a conditioned medium derived from confluent HEK293 cells and HEK293 cells in which LOX and the various LOX mutants were expressed. Cleaved LOX is indicated by an arrow.

**Figure 6 ijms-23-11322-f006:**
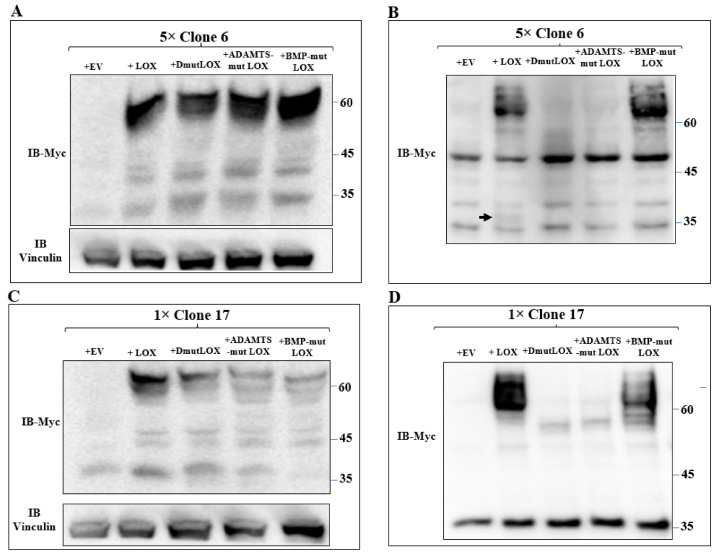
Expression of the three point-mutated LOX variants in 5× and 1× knock-out MB-MDA-231 cells. (**A**,**C**) Western blot analysis using anti-myc antibodies of myc-tagged LOX and the myc-tagged, point-mutated LOX variants in cell lysates or (**B**,**D**) in conditioned media, derived from clone 6 5× knock-out MDA-MB-231 cells infected with either empty expression vectors (EVs) or the cDNAs encoding the indicated LOX variants (**A**,**B**). Similar analysis is shown for clone 17 1× knock-out MDA-MB-231 cells that were infected with the same constructs (**C**,**D**). Cleaved LOX is indicated by an arrow.

**Figure 7 ijms-23-11322-f007:**
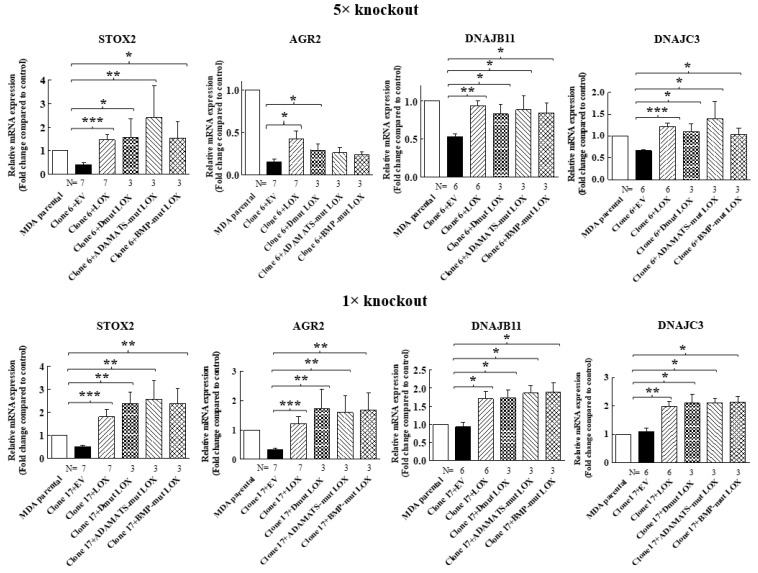
Point-mutated LOX variants that fail to be cleaved are still able to promote the expression of LOX-regulated genes. The expression levels of the STOX2, AGR2, DNAJB11 and DNAJC3 mRNA were determined in MB-MDA-231 as well as in 5× and 1× knock-out cells using qRT-PCR. The expression was compared with the expression level of these genes following the expression of the indicated LOX variants. Results were normalized with the expression level in MB-MDA-231 parental cells. Data represent the mean ± SEM of N independent experiments. Statistical significance was evaluated using the Mann–Whitney test. * *p* < 0.05, ** *p* < 0.01, *** *p* < 0.001.

**Figure 8 ijms-23-11322-f008:**
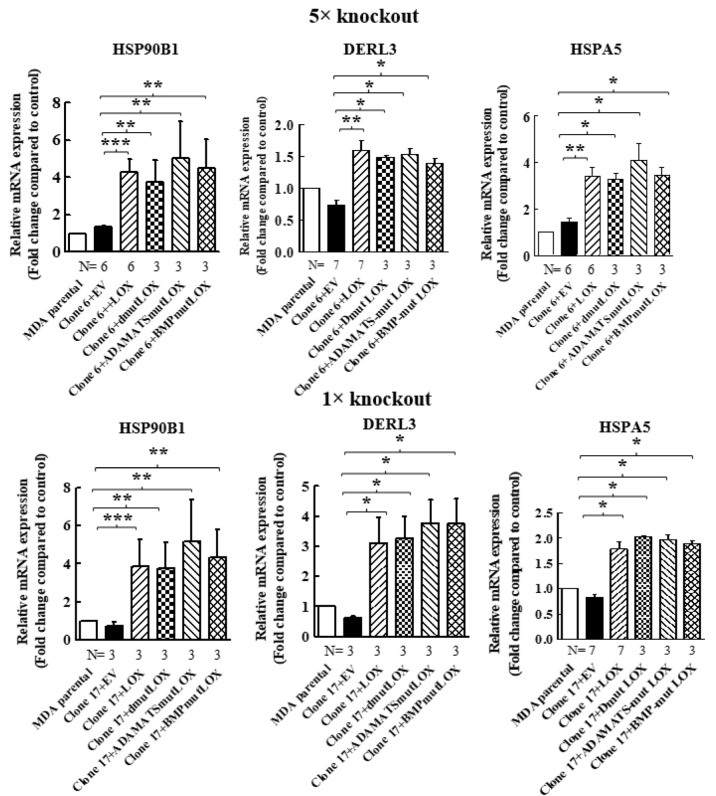
Point-mutated LOX variants that fail to be cleaved are also able to promote the expression of genes that were only upregulated following LOX expression in 5× and 1× knock-out cells. The expression levels of the HSP90B1, DERL3 and HSPA5 mRNAs were determined in MB-MDA-231 as well as in 5× and 1× knock-out cells using qRT-PCR. The expression was compared with the expression levels of these genes following the expression of the indicated LOX variants. Results were normalized with the expression level in MB-MDA-231 parental cells. Data represent the mean ± SEM of N independent experiments. Statistical significance was evaluated using the Mann–Whitney test. * *p* < 0.05, ** *p* < 0.01, *** *p* < 0.001.

**Figure 9 ijms-23-11322-f009:**
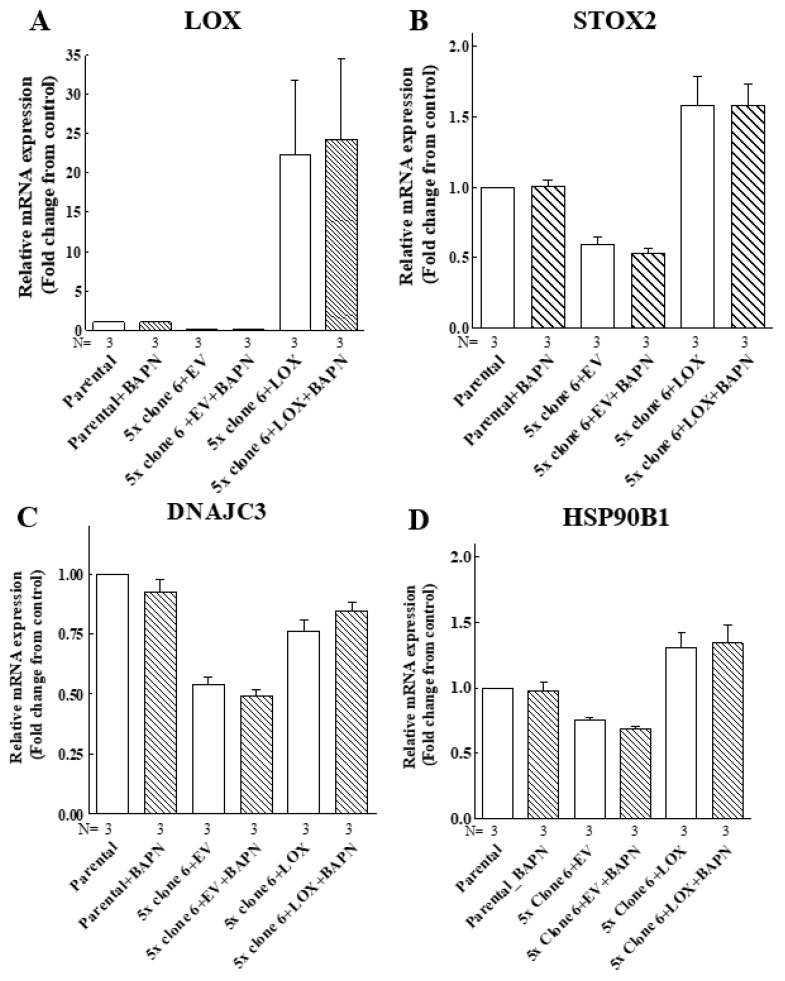
The LOX-induced expression of the STOX2, DNAJC3 and HSP90B1 genes does not require LOX enzyme activity. Parental MB-MDA-231 cells (parental), clone 6 5× knock-out cells infected with empty lentiviruses (5× clone 6 + EV) or clone 6 x5 knock-out cells expressing the LOX cDNA (5× clone 6 + LOX) were cultured in the absence or presence of 250 μM BAPN and 5 μM CuSO_4_ for 48 h. Subsequently, the mRNA expression of LOX (**A**), STOX2 (**B**), DNAJC3 (**C**) and HSP90B1 (**D**) was examined using qRT-PCR. Results were normalized with the expression level in MB-MDA-231 parental cells in each of the experiments. Data represent the mean ± SEM of N independent experiments.

## Data Availability

Data is available upon request from the corresponding author.

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
