# Peer review of "Knock-Out of the Five Lysyl-Oxidase Family Genes Enables Identification of Lysyl-Oxidase Pro-Enzyme Regulated Genes"

_ijms, 2022, doi:10.3390/ijms231911322_

Round 1

Reviewer 1 Report (Previous Reviewer 1)

Accept

Author Response

-

Reviewer 2 Report (Previous Reviewer 2)

General points

This is the ambitious research that quests biological roles of the lysyl oxidase (LOX) enzyme activity on regulations of cellular functions by knocking-out all the 5 family genes of LOX.

This manuscript is the revised version and this is my second-time for reviewing this study.

I saw this with a great interest for the improvement on issues I had pointed. 

Briefly, it was a little bit disappointing, because there were few mentions about my previous points concerns with the experimental procedure such as historical presence or absence of studies using similar experimental procedure, and author's thoughts about the increase of experimental parameters by employing this knockout method.

Each point

Title

The title only explains the experimental procedure, without presenting any biological / physiological findings in this study.

I feel this is not appropriate for this journal, because I understand this journal's scope is biological / physiological discussion, but not about technical development.

Previous version of the title "The lysyl-oxidase pro-enzyme regulates gene expression in breast cancer cells" was also lack of aspect of biology / physiology.

By seeing this kind of expressions twice, I am now suspicious that since even the authors are not confident enough in their experimental validity, they do not state clearly about biological / physiological significance extracted from what they observed.

Introduction

Again, I greatly appreciate their challenge on this experimental procedure, because it is so ambitious and interesting.

However, I do not feel whether the procedure is valid or not is assessed enough.

At least the presence or absence of the similar procedure that knocking out all the family genes should be described in the Introduction, and if there were no examples, then "this is the first, and we tried" should be noticed.

If there are some examples, the validity in employing this procedure should be mentioned in comparison with preceding examples.

In either case, possible technical / biological issues emerge by employing this procedure should be mentioned in balancing with descriptions in the Results and the Discussions.

Result / Discussion

As mentioned above, the validity of the procedure should at least be discussed, moreover, experiments proving the validity should be done if necessary / possible.

Particularly on the order of knockout, current version of description stands on the classification of LOX subfamily, but I think it is insufficient.

If they think the order based on the classification is proper, it is possible to prove by changing the order oppositely.

I feel the authors are, as if, refusing to assess their experimental validity because I do not catch that they have these points of view not only from previous version but also in current version of the manuscript, again.

Please notice that I do not say that the experiment above is essential.

I point and ask to describe clearly for the readers that this study includes the issues above, and the data as well as the discussions are together with these issues.

Round 2

Reviewer 2 Report (Previous Reviewer 2)

Multiple knockdown was executed in this study, and there are a few similar experimental examples as the author points.

Thus, this reviewer greatly appreciates this ambitious attitude of the authors and basically willing to accept this manuscript. 

For this aim, I had mentioned on some points which should help proper presentation of features, results, significance and limitations of this study toward readers.

Since we got a great improvement in this 2nd-time reviewing, I would like to make decision of acceptance.

However, it is disappointing that still some points were not been understood by the author.

I had pointed out that since there are few examples of this strategy, the fact itself should be described, and the reader should be informed the differences (sequential or simultaneous), advantages and/or disadvantages and so on between the examples.

As multiple KOs were performed sequentially in this study, clones were taken each time. Those clones should have changed traits at each KO, and the changes should be further compounded as the KOs are repeated.

Although LOX family proteins are thought to have the same enzymatic activity, as the authors point out, they can be structurally divided into subfamilies, which may lead to differences in the distribution and regulation of enzymatic activity in cells as well as tissues, and therefore, depending on which family proteins are KO'd, the resulting transformational changes are likely to go beyond the loss of enzyme activity.

Based on this recognition, the order of KOs is very important in a strategy such as the one used in this study, and I was very dissatisfied with the lack of mention to this point.

However, when I said that I did not request experiments in which the order of KOs was changed, I meant that it would be kind enough for the reader and appropriate as a self-evaluation of the study to note that the authors are aware of the above possibilities, that they exist for consideration in this study, and that the results of gene expression changes obtained are together with that.

It was sufficient to state that, for example, it is undeniable that the obtained changes in gene expression may vary depending on the order of KOs, and it is necessary to confirm this by experimentation - this is the "validation" of the strategy that I pointed out as necessary but did not been understood -, we would like to consider this in the future, and the order employed this time is as the first step of this study.

If the authors stand on this understanding, I believe that it would have been appropriate and possible to use a title mentions "possible" relationship between enzyme activity and changes in gene expression rather than assertive expressions.

It is very disappointing that this point was not been understood and that the reply declaring they do not understand what the reviewer is talking about from beginning to end.

This manuscript is a resubmission of an earlier submission. The following is a list of the peer review reports and author responses from that submission.

Round 1

Reviewer 1 Report

The article is very well written and the research and well supported by the conclusions. The paper may be accepted in its current form after removing grammatical mistakes.

Reviewer 2 Report

General points

Knocking out (KO) all the family member is an interesting and radical procedure.

To know how the enzyme family bearing common enzymatic activities but is assumed have different distributions and / or regulation mechanism act differently, this kind of radical way may be a strong and useful, however simultaneously, increase of parameters by which this procedure invites may require very much careful analyses with originality as well as ingenuity.

Each point

1. Preparation of manuscript

Are the fonts constant? (for example, line 31-33)

What is {18399, 23410, 18849} in line 41?

Including these points, I am curious whether this manuscript is prepared carefully enough?

Please check entirely again.

2. Title

I think current title is too much obscure.

How does LOX pro-enzyme regulate what factors, and what it means should be expressed concisely.

I do not feel novelty in the current form of the title, and moreover, the importance of this work may not appeal at all to the readers who yet have interest in LOX family genes.

3. Experimental procedures

I greatly appreciate for the concept of complete knockout of sole family genes, because I don't know other examples.

However, as I pointed in the General points above, requirement for carefulness and ingenuity are easily expected because of the greater increase of parameters concerned.

I think the presence or absence of similar examples, the comparison between present analyses, and whether it is appropriate enough or not should be discussed, at least in the Discussion section.

I also think that the explanation about the idea and/or concept as well as advantage and problems in the Introduction section, or experimental item to verifying the procedure in the Results section should be also possible or necessary.

I feel this point is not enough in this manuscript because I don't see it as far as I checked (I am so sorry if I miss it).

I am also curious about the order of KO. Of course, current data is OK, but whether this way is the best or not should be assessed somehow or at least discussed.

Do the authors think that the results obtained this time is same if you do the KO in different order?

If the order affects somehow to the results, emergence of the further experimental development including simultaneous KO is concerned, and also whether MDA-MB-231 cell was suitable or not to explore in this kind of ambitious experiment can be emerged in this context.

Of course, I don't ask to conclude this kind of thing in this manuscript, however, whether the authors are aware of, and whether the experimental design are considered for this kind of thing should be mentioned somehow.

4.

In figure 3, four genes of STOX2, AGR2, DNAJC3, DNAJB11 were downregulated by the 5X KO and were rescued by only LOX but not by other family genes, and authors conclude that these 4 genes are under control of LOX. However, DNAJC3 among them was not downregulated by the LOX 1x KO while upregulated by the re-expression of LOX.

In ordinal 1x KO experiment, this result may lead to the suggestion that DNAJC3 is not locate under LOX at least directly, and the upregulation by the LOX re-expression should have some arguments, for example artificial or at least non-physiological.

I think this kind of thing should be the very problem in the 5x KO experiment and I point in points #3.

I guess there should be similar issues through this manuscript.

In total, I greatly appreciate this manuscript in the aspect of the ambitious procedure and data obtained by this procedure, however simultaneously, got the impression of immaturity in appropriate processing of what they obtained from the experiments.

In other words, I feel problems in the preparation of manuscript rather than any deficiency or failure in experiment or analyses.

I would like to ask for the re-construction of the manuscript.

Reviewer 3 Report

Review of:

The lysyl-oxidase pro-enzyme regulates gene expression in breast cancer cells

Comments to the Authors

General remarks:

The authors investigate the role of LOX in regulating gene expression. They use two knockout models in MDA-MB-231 breast cancer cells with either all five LOX genes or only LOX knocked out. Then they overexpress the LOX gene in this background. The knockouts are done by Crispr and the overexpression is done with a CMV promoter.

The authors claim the 4 genes (AGR2, STOX2, DNAJB11 and DNAJC3) are downregulated in the knockouts and upregulated when LOX is overexpressed. Another set of genes (HSP90B1, DERL3 and HSPA5) are only upregulated when LOX is overexpressed.

The gene regulation is independent on the cleavage of LOX by the proteases BMP-1 or ADAMTS2/14 as shown by mutation or an inhibitor.

The observation by the authors does not directly link LOX with gene regulation. It could be merely coincidental. I am missing a functional link between LOX and the gene regulation. It could be simply an effect of overexpressing a gene at excess which initiates the cell’s stress response. The CMV promoter is an extremely strong promoter causing overexpression to a non-physiological range. This can be seen in figures 2A and B: while endogenous LOX in the parental is a very faint band, overexpressed LOX is extremely abundant. A Coomassie gel or Ponceau red staining may reveal how much of the total cellular protein is LOX.

Also, the regulated genes are stress response proteins involved in heat shock, protein folding regulated by TGF-b. These genes may be induced when any gene is overexpressed causing stress to the cell.

Additionally, the observation that the functional cleavage of LOX does not impact the gene regulation points to the fact that it is just overexpression causing the upregulation of the stress genes.

A more modulated range of expression within physiological range may give a different gene-regulation readout.

 The authors need to prove that the link between LOX and regulated genes is not coincidental but incidental. This could be done with the KO models rather than with the non-physiological overexpression.

I am also missing more functional assays: the invasion assay in figure 1E is a fantastic readout for the effect the various knockouts have. That should be quantified. How does a modulation of overexpression change the invasion? Would manipulating the expression of the target genes change the invasion?

Another readout could be cell proliferation/apoptosis assays. The authors mention hypoxia as a physiologic regulator of LOX, does that change the expression of the target genes?

The authors published functional assays using LOX, maybe some of those can be used as a readout for the gene regulation.

Do the authors hypothesize LOX to be a transcription factor? Then a chipseq experiment may prove a direct interaction with enhancer regions within the target genes.

Specific comments:

Line 4: “nd” needs to be replaced by “and”

Lines 14, 45: Human gene symbols are all capitalized

Line 41: References are not formatted correctly

Line 161: It’s RNA sequencing not RNS

Line 214: It’s Fig. 1A

Line 216: It’s Fig. 1B

Figure 1C: adding the translated protein sequence would illustrate the frameshift better

Figure 2C: Adding clone 17 1x would be informative

Figure 5B, 6: It’s not clear to me which band if any at all corresponds to cleaved LOX, replacing the arrow with a box would clarify which band to look at.

Some figure legends: the exact number of replicates should be given.

It would be great if the authors would deposit the RNAseq data to the GEO database to give other researchers access to their data.

Round 2

Reviewer 3 Report

The authors resubmitted the manuscript without any new experiments as they considered all my suggestions as "out of the scope of this manuscript".

While they agree mostly with my criticism, the only substantial change is the wording in the title and the addition of an explanatory paragraph to the discussion.

The reason I reviewed the manuscript to improve the quality of the research being done and put it into the context of existing data. Undoubtedly, the research the authors have been done is important and was done in a thoughtful manner. I think there is great potential if functional assays were added. The authors have a great assay in hand: the invasion assay. Also proliferation and apoptosis assays could be done. The authors mention hypoxia as regulating STOX2. The authors need to link LOX with at least one gene (e.g. STOX2, since this is the only non-stress response gene).

Just observing that the targeted genes were down regulated in the KO and upregulated in the overexpressor I think is not enough.